# Maximizing Spatio-Temporal Entropy of Deep 3D CNNs for Efficient Video Recognition

**Junyan Wang**[1][§]**, Zhenhong Sun**[12][§]**, Yichen Qian**[2]**, Dong Gong**[1]**, Xiuyu Sun**[2][*]**Ming Lin**[3][‡]**,
Maurice Pagnucco**[1]**, Yang Song**[1]
[1]University of New South Wales
[2]DAMO Academy, Alibaba Group
[3]Amazon
{junyan.wang, dong.gong, yang.song1}@unsw.edu.au
{zhenhong.szh, yichen.qyc, xiuyu.sxy}@alibaba-inc.com
{morri}@cse.unsw.edu.au, {minglamz}@amazon.com

## Abstract

3D convolution neural networks (CNNs) have been the prevailing option for video recognition. To capture the temporal information, 3D convolutions are computed along the sequences, leading to cubically growing and expensive computations. To reduce the computational cost, previous methods resort to manually designed 3D/2D CNN structures with approximations or automatic search, which sacrifice the modeling ability or make training time-consuming. In this work, we propose to automatically design efficient 3D CNN architectures via a novel training-free neural architecture search approach tailored for 3D CNNs considering the model complexity. To measure the expressiveness of 3D CNNs efficiently, we formulate a 3D CNN as an information system and derive an analytic entropy score, based on the Maximum Entropy Principle. Specifically, we propose a spatio-temporal entropy score (STEntr-Score) with a refinement factor to handle the discrepancy of visual information in spatial and temporal dimensions, through dynamically leveraging the correlation between the feature map size and kernel size depth-wisely. Highly efficient and expressive 3D CNN architectures, *i.e.,* entropy-based 3D CNNs (E3D family), can then be efficiently searched by maximizing the STEntr-Score under a given computational budget, via an evolutionary algorithm without training the network parameters. Extensive experiments on Something-Something V1&V2 and Kinetics400 demonstrate that the E3D family achieves state-of-the-art performance with higher computational efficiency. Code is available at https://github.com/alibaba/lightweight-neural-architecture-search.

## 1 Introduction

Video recognition is a fundamental task for video understanding. To capture the visual information in both temporal and spatial domains from high-quality large-scale videos, most works have been focusing on proposing highly expressive models which, however, lead to higher computational costs Kondratyuk et al. (2021); Zhang et al. (2022); Li et al.. Recent research shows that 3D CNNs achieve excellent performance on large-scale benchmarks (Hara et al., 2018) with unified computations to capture spatio-temporal features jointly. However, the computational cost grows cubically in standard 3D convolution, making it prohibitive for high-resolution long-duration videos. Previous works propose to improve the efficiency of 3D CNNs via 2D decomposition or approximation manually (Carreira & Zisserman, 2017; Tran et al., 2018; Feichtenhofer, 2020). Some practices have also been conducted to manually design efficient 3D CNNs relying on heuristics or experiences (Hara et al., 2018; Feichtenhofer, 2020). The manually designed 3D or 2D CNN structures cost massive efforts and time in strengthening the modeling ability. Neural Architecture Search (NAS) approaches (Kondratyuk et al., 2021; Wang et al., 2020) can automatically generate 3D CNN architectures with

---

*Corresponding author, [§]equal contribution, [‡]work done in Alibaba.

higher modeling ability. However, searching for a single 3D architecture requires days on multiple GPUs or TPUs, as training and evaluation of the accuracy indicator are required in the process, making the automatic 3D CNN design process time-consuming and/or hardware-dependent.

To tackle the above issues, we study how to automatically generate (or design) efficient and expressive 3D CNNs with limited computations. Recently, training-free technologies have been introduced by some approaches (Chen et al., 2021; Lin et al., 2021; Sun et al., 2022b), in which kernel spectrum analysis or forward inference are adopted to measure the expressiveness of spatial 2D CNNs. Inspired by the training-free concept and information theory, we suggest that a deep network can be regarded as an information system, and measuring the expressiveness of the network can be considered equivalent to analyzing how much information it can capture. Therefore, based on the **Maximum Entropy Principle** (Jaynes, 1957), the probability distribution of the system that best represents the current state of knowledge is the one with the highest entropy. However, as discussed in (Xie et al., 2018), the information in spatial and temporal domains is different in natural video data. The spatial dimension is usually limited to some local properties, like connectivity (Claramunt, 2012), while the temporal dimension usually contains more drastic variations with more complex information. To address the spatio-temporal discrepancy in video data, we conduct a kernel selection experiment and observe that different 3D kernel selections in different stages have different effects on performance, and the focus of 3D CNNs changes from spatial information to spatio-temporal information, as the network depth increases. We thus consider that the design of 3D CNN architecture should focus on spatial-temporal aggregation depth-wisely.

The above analysis has motivated us to propose a training-free NAS approach to obtain optimal architectures, *i.e.,* entropy-based 3D CNNs (**E3D** family). Concretely, we first formulate a 3D CNN-based architecture as an information system whose expressiveness can be measured by the value of its differential entropy. We then derive the upper bound of the differential entropy using an analytic formulation, named **Spatio-Temporal Entropy Score** (STEntr-Score), conditioned on spatio-temporal aggregation by dynamically measuring the correlation between feature map size and kernel size depth-wisely. Finally, an evolutionary algorithm is employed to identify the optimal architecture utilizing the STEntr-Score without training network parameters during searching. In summary, the key contributions of our work are as follows:
• We present a novel training-free neural architecture search approach to design efficient 3D CNN architectures. Instead of using forward inference estimation, we calculate the differential entropy of a 3D CNN by an analytic formulation under Maximum Entropy Principle.
• We investigate the video data characteristics in spatial and temporal domains and correlation between feature map with kernel selection, then propose the corresponding spatio-temporal entropy score to estimate the spatio-temporal aggregation dynamically, with a spatio-temporal refinement mechanism to handle the information discrepancy.
• Each model of E3D family can be searched within three hours on a desktop CPU, and the models demonstrate state-of-the-art performance on various video recognition datasets.

## 2 RELATED WORK

**Action recognition.** 2D CNNs lack temporal modeling for video sequences, and many approaches (Wang et al., 2016; Lin et al., 2019; Li et al., 2020; Wang et al., 2021a;b; Huang et al., 2022) focused on designing an extended module for temporal information learning. Meanwhile, 3D CNN-based frameworks have a spatio-temporal modeling capability, which improves model performance for video action recognition (Tran et al., 2015; Carreira & Zisserman, 2017; Feichtenhofer, 2020; Kondratyuk et al., 2021). Some attempts (Feichtenhofer, 2020; Fan et al., 2020; Kondratyuk et al., 2021) focused on designing efficient 3D CNN-based architectures. For example, X3D (Feichtenhofer, 2020) progressively expands a tiny 2D image classification architecture along multiple network axes, in space, time, width and depth. Our work also focuses on designing efficient 3D CNN-based architectures, but in a deterministic manner with entropy-based information criterion analysis.

**Maximum Entropy Principle**. The Principle of Maximum Entropy is one of the fundamental principles in Physics and Information Theory (Shannon, 1948; Reza, 1994; Kullback, 1997; Brillouin, 2013). Accompanied by the widespread applications of deep learning, many theoretical studies (Saxe et al., 2019; Chan et al., 2021; Yu et al., 2020; Sun et al., 2022b) try to understand the success

of deep learning based on the Maximum Entropy Principle. Our work focuses on video recognition and explores the aggregation of spatio-temporal information under the Maximum Entropy Principle.

**Training-Free NAS**. To reduce the search time of NAS, recent attempts (Mellor et al., 2021; Chen et al., 2021; Tanaka et al., 2020; Lin et al., 2021; Sun et al., 2022a;b; Zhou et al., 2022; Chen et al., 2022; Lin et al., 2020) proposed training-free strategies for architecture searching, which construct an alternative score to rank the initialized networks without training. For example, the work of (Sun et al., 2022b) maximizes the differential entropy of detection backbones, leading to a better feature extractor for object detection under the given computational budgets. However, these methods construct scores on spatial 2D CNNs, and cannot handle the discrepancy of visual information in spatial and temporal dimensions of 3D CNNs. In order to address the above issues, our work aims to optimize the network architecture by considering spatio-temporal dimensions aggregation.

## 3 THE PROPOSED APPROACH

In this section, we first present a detailed technical description of the derivation process of an analytical solution and propose the STEntr-Score with a refinement factor to handle the discrepancy of visual information in spatial and temporal dimensions. Then we give an overview of the search strategy for the E3D family, via an evolutionary algorithm without training the network parameters.

### 3.1 PRELIMINARY

In *Information Theory*, differential entropy is employed to represent the information capacity of an information system by measuring the output of the system (Jaynes, 1957; Reza, 1994; Kullback, 1997; Brillouin, 2013; Norwich, 1993). Generally, the output of a system is a high-dimensional continuous variable with a complex probability distribution, making it difficult to compute the precise value of its entropy directly. Based on the **Maximum Entropy Principle** (Jaynes, 1957), a common alternative approach is to estimate the upper bound of the entropy (Cover & Thomas, 2012), as:

**Theorem 1** *For any continuous distribution $P(x)$ of mean $\mu$ and variance $\sigma^2$, its differential entropy is maximized when $P(x)$ is a Gaussian distribution $\mathcal{N}(\mu, \sigma^2)$.*

Thus, the differential entropy of any distribution is upper bound by the Gaussian distribution with the same mean and variance. Suppose $x$ is sampled from Gaussian distribution $\mathcal{N}(\mu, \sigma^2)$, the differential entropy (Norwich, 1993) of $x$ is then:

$$\mathcal{H}(x) = \int_{-\infty}^{+\infty} -log(P(x))P(x)dx \propto log(\sigma^2), \tag{1}$$

where $P(x)$ represents the probability density function of $x$. Note that the entropy of the Gaussian distribution depends only on the variance $\sigma^2$, and a simple proof is included in the **Appendix B.1**.

According to successful deep learning applications (Saxe et al., 2019; Chan et al., 2021; Yu et al., 2020; Sun et al., 2022b;a) of Maximum Entropy Principle, a deep neural network can be regarded as an information system, and the differential entropy of the last output feature map represents the expressiveness of the system. Recent method (Sun et al., 2022b) estimates the entropy of 2D CNNs by simply computing the feature map variance via sampling input data and initializing network parameters from a random standard Gaussian distribution. However, when migrating to 3D CNNs, how to efficiently reduce the random sampling noise due to the random initialization, and how to estimate the entropy after aggregating spatial and temporal dimensions in 3D CNNs design, still remain open questions. We then propose our method to address these problems.

### 3.2 STATISTICAL ANALYSIS OF ENTROPY IN DEEP 3D CNNS

**Simple Network Space**. Following the idea that *simpler is better* (Lin et al., 2021; Sun et al., 2022b), we apply vanilla 3D CNNs without considering auxiliary modules (*e.g.,* BN (Ioffe & Szegedy, 2015), Reslink (He et al., 2016), SE block (Hu et al., 2018) and so on) to conduct analysis of network architectures. Formally, given a convolutional network with $L$ layers of weights $\boldsymbol{W}^1, ..., \boldsymbol{W}^L$, the forward inference with a simple network space is given by:

$$\boldsymbol{x}^l = \boldsymbol{W}^l * \boldsymbol{x}^{l-1} \quad \text{for } l = 1, \ldots, L, \tag{2}$$

where $x^l$ denotes the $l^{th}$ layer feature map. For holistic analysis, the bias of the convolutional layer is set to zero and the activation function is omitted in the network for simplification. Auxiliary modules are ignored during entropy calculation and plugged into the backbone without special modification during training. A detailed discussion about these rules is included in **Appendix C**.

Since the input data and network parameters are randomly sampled from Gaussian distributions, the forward entropy calculation will be inconsistent, which might lead to random sampling noise. To obtain a valid entropy value, computing an average value from multiple sampling iterations and increasing the value of batch size or resolution can be adopted to reduce the noise. These operations are however time-consuming and cost higher computational resources. To this end, we propose to explore the statistical characteristics of the forward inference, to provide an efficient solution.

**Maximum Entropy of 3D CNNs**. We first consider the *product law of expectation* (Mood, 1950) and the *Bienaymé's identity* in probability theory (Loeve, 2017), as follows:

**Theorem 2** *Given two independent random variables $v_1$, $v_2$, the expectation of their product $v_1 v_2$ is:* $\mathbb{E}(v_1 v_2) = \mathbb{E}(v_1)\mathbb{E}(v_2)$.

**Theorem 3** *Given $n$ random variables $\{v_1, v_2, ..., v_i, v_{i+1}, ..., v_n\}$ which are pairwise independent integrable, the sums of their expectations and variances are:* $\mathbb{E}(\sum_{i=1}^{n} v_i) = \sum_{i=1}^{n} \mathbb{E}(v_i)$, *and* $\mathbb{D}^2(\sum_{i=1}^{n} v_i) = \sum_{i=1}^{n} \mathbb{D}^2(v_i)$.

We can thus compute the expectation and variance of $l^{th}$ layer feature map element $\boldsymbol{x}_i^l$ as:

$$\mathbb{E}(\boldsymbol{x}_i^l) = \sum_{t=1}^{K_t^l} \sum_{h=1}^{K_h^l} \sum_{w=1}^{K_w^l} \sum_{c=1}^{C^{l-1}} \left[ \mathbb{E}(\boldsymbol{x}_{cthw}^{l-1})\mathbb{E}(\boldsymbol{W}_{cthw}^l) \right], \tag{3}$$

$$\mathbb{D}^2(\boldsymbol{x}_i^l) = \sum_{t=1}^{K_t^l} \sum_{h=1}^{K_h^l} \sum_{w=1}^{K_w^l} \sum_{c=1}^{C^{l-1}} \left\{ \mathbb{D}^2(\boldsymbol{x}_{cthw}^{l-1})\mathbb{D}^2(\boldsymbol{W}_{cthw}^l) + \mathbb{D}^2(\boldsymbol{x}_{cthw}^{l-1})\left[\mathbb{E}(\boldsymbol{W}_{cthw}^l)\right]^2 \right. $$
$$\left. + \mathbb{D}^2(\boldsymbol{W}_{cthw}^l)\left[\mathbb{E}(\boldsymbol{x}_{cthw}^{l-1})\right]^2 \right\}, \tag{4}$$

where $\{K_t^l, K_h^l, K_w^l\}$ represents the kernel size of the $l^{th}$ layer in the 3D CNN, and $C^{l-1}$ denotes its input channels size. Note that $C^{l-1}$ is equal to 1 when the layer is a depth-wise convolution. Besides, $t, h, w$ denote the temporal, height, and width positions, respectively. A simple proof is included in **Appendix B.2**.

The input $x^0$ is initialized from a standard Gaussian distribution, which means that its expectation $\mathbb{E}(x^0) = 0$ and variance $\mathbb{D}^2(x^0) = 1$. From the perspective of statistics, we can regard $\mathbb{D}^2(x_{cthw}^0) = 1$ when sampling sufficient times. Also, suppose that all parameters are initialized from a zero-mean Gaussian distribution, and thus the variance of the last layer $\mathbb{D}^2(x^L)$ can be computed by propagating the variances from previous layers as:

$$\mathbb{D}^2(\boldsymbol{x}_i^L) = \prod_{l=1}^{L} K_t^l K_h^l K_w^l C^{l-1} \mathbb{D}^2(\boldsymbol{W}_{cthw}^l). \tag{5}$$

Finally, by combining Eq. (5) and Eq. (1), we derive that the upper bound entropy is numerically proportional to:

$$\mathcal{H}(F) \propto \sum_{l=1}^{L} log(K_t^l K_h^l K_w^l C^{l-1} \mathbb{D}^2(\boldsymbol{W}_{cthw}^l)), \tag{6}$$

where detailed proof is included in **Appendix B.3**. By assuming that the parameters of each layer are initialized with a standard Gaussian distribution with $\mathbb{D}(\boldsymbol{W}_{cthw}^l) = 1$, the entropy score defined in Eq. (6) can be written as $\sum_{l=1}^{L} log(K_t^l K_h^l K_w^l C^{l-1})$. It measures the influence of kernel size and channel dimension on the entropy score in a homogeneous way, named **HomoEntr-Score**. This analytic formulation does not require random sampling, thus no random sampling noise exists.

### 3.3 SPATIO-TEMPORAL ENTROPY SCORE

The HomoEntr-Score is derived from the analysis with an independent and identical assumption on the input elements (and the corresponding intermediate features). Although it can generally represent the expressiveness characteristics of a neural network, there is a gap between HomoEntr-Score and reality on 3D CNNs. When directly applying it on 3D CNNs for handling video sequences, we realize the HomoEntr-Score with the independent and identical assumption cannot capture the discrepancy of the visual information in the spatial and temporal domain, as the information between spatial and temporal dimensions in video data is different in video recognition. The gap leads to some issues with HomoEntr-Score for modeling video data with 3D CNNs. The observations will be discussed and analyzed in the following. Note that HomoEntr-Score (and similar approaches (Sun et al., 2022b)) can work well for modeling the expressiveness of 2D CNNs since there is no (obvious) discrepancy on the information of the two directions in 2D images statistically. Based on the analyses, we propose a Spatio-Temporal Entropy Score (STEntr-Score) for 3D CNNs on video data, where a Spatio-temporal refinement factor is introduced to handle the information discrepancy.

| Model | Top1 | Params (M) | FLOPs (G) | HomoEntr Score | | Model | Top1 | Params (M) | FLOPs (G) | HomoEntr Score |
|-------|------|-----------|-----------|----------------|---|-------|------|-----------|-----------|----------------|
| S-2 | **45.15%** | 3.33 | 1.93 | 178.5 | | T-2 | 41.59% | 3.32 | 1.93 | 177.7 |
| S-3 | 44.87% | 3.33 | 1.93 | 178.4 | | T-3 | 42.93% | 3.32 | 1.93 | 177.8 |
| S-4 | 44.35% | 3.33 | 1.94 | 178.4 | | T-4 | 43.17% | 3.32 | 1.92 | 177.8 |
| S-5 | 43.85% | 3.33 | 1.94 | 178.4 | | T-5 | **43.43%** | 3.32 | 1.92 | 177.9 |
| S-6 | 43.83% | 3.33 | 1.94 | 178.4 | | T-6 | 43.35% | 3.32 | 1.92 | 177.9 |

Table 1: Results of different kernel positions on the Sth-Sth V1 validation dataset. All model structures are based on X3D-S (Feichtenhofer, 2020). "S-N" models mean only stage N selects 1×5×5 kernel, and others select 3×3×3. "T-N" models mean only stage N selects 3×3×3 kernel, and others select 1×5×5. Note that we divide stage 4 of X3D with 11 layers into two stages (5 and 6 layers).

**Kernel Selection Observations**. We conduct an experiment to explore how different 3D convolutional kernel sizes at different stages (*i.e.,* layer blocks at different positions in the network) impact the performance, as shown in Table 1. All models are based on X3D-S but with different kernels in different stages. We set 1×5×5 and 3×3×3 kernels at the different stages in the 3D CNNs, which are typical 3D convolutional kernels for learning spatio-temporal information. These two different choices enable a layer to aggregate the visual information focusing on different spatial and temporal dimensions, with the receptive field of CNN in the most pertinent directions. In Table 1, the performances of S-2 and S-3 models are higher than X3D-S with only 3×3×3 kernels (44.6% in Table 2), and S-series outperform T-series, which show that kernel selection at different stages influences the performance significantly, and that different stages may prefer different kernel sizes, respectively. Although the kernel selections (with different spatio-temporal dimensions) at different stages lead to different effects on performance, the corresponding 3D CNNs have similar HomoEntr-Score.

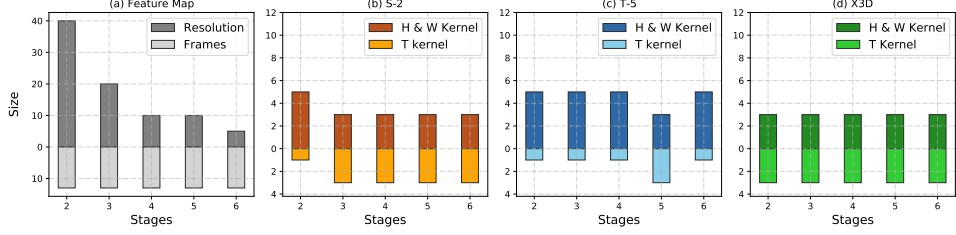

Figure 1: Input feature map size and kernel sizes of S-2, T-5 and X3D model in each stage.

According to the downsampling strategy of the 3D CNNs, spatial resolutions become smaller from the large input as the depth increases, while the temporal frame size remains a certain value, as shown in Figure 1a. Through analyzing the results in Table 1, we can infer that spatial kernels (like 1×5×5) can obtain spatial information more effectively at low-level stages, and spatio-temporal kernels (like 3×3×3) are more stable to obtain spatio-temporal information at high-level stages. Meanwhile, the similarity between the input feature map and the kernel size of the S-2 model at each stage is higher than that of the T-5 model or X3D, according to Figure 1. We thus consider that

with the higher correlation between the feature map size and kernel size depth-wisely, the model can obtain higher expressiveness of spatial and temporal information.

**Spatio-Temporal Refinement**. To estimate the correlation between feature map and kernel size in different depths, we first define two vectors: the input feature map size $\boldsymbol{S} = [T, H, W]$ and the 3D kernel size $\boldsymbol{K} = [K_t, K_h, K_w]$ in a convolutional layer, where $\{T, H, W\} \in \mathbb{R}$ represent frame, height and width dimension size. We compute the distance $\hat{\mathcal{D}}$ based on commonly used cosine distance as:

$$\hat{\mathcal{D}}(\boldsymbol{S}, \boldsymbol{K}) = -log(\mathcal{D}_{cosine}(\boldsymbol{S}, \boldsymbol{K})) = -log\big(1 - \frac{\boldsymbol{S} \cdot \boldsymbol{K}}{\|\boldsymbol{S}\|\|\boldsymbol{K}\|}\big) \,, \tag{7}$$

where $\mathcal{D}_{cosine}$ represents the *Cosine Distance* function, and we expand the diversity of the cosine distance by using $log$. We thus utilize the distance $\hat{\mathcal{D}}$ between $\boldsymbol{S}$ and $\boldsymbol{K}$ in each layer to define the variance of weight dynamically. Finally, we refine the upper bound differential entropy as:

$$\mathcal{H}(F) \propto \sum_{l=1}^{L} log(K_t^l K_h^l K_w^l C^{l-1} \cdot \hat{\mathcal{D}}(\boldsymbol{S}^l, \boldsymbol{K}^l)) \,. \tag{8}$$

We name this analytic formulation of Eq. (8) as **Spatio-Temporal Entropy Score** (STEntr-Score) to measure the aggregation of spatio-temporal dimensions. After spatio-temporal refinement, we recalculate the entropy of each model by STEntr-Score in Table 1, and present the relationship between accuracy with STEntr-Score and HomoEntr-Score in Figure 2. According to this figure, STEntr-Score is positively correlated with Top1 accuracy which indicates that the proposed spatio-temporal refinement can handle the discrepancy of visual information in spatial and temporal dimensions.

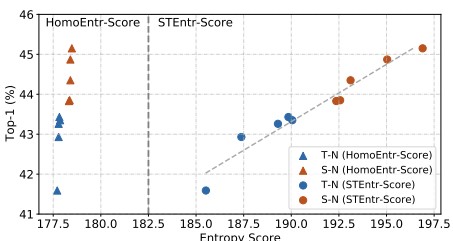

Figure 2: Top-1 accuracy vs. STEntr-Score and HomoEntr-Score.

### 3.4 3D CNN SEARCHING STRATEGY

Utilizing STEntr-Score, we apply the basic **Evolutionary Algorithm** (EA) to find the optimal 3D CNN architectures, which is similar to (Lin et al., 2021; Sun et al., 2022b). We initialize a population of candidates randomly under a small budget and define the 3D kernel search space within each layer with two options: $\{1 \times (k^{space})^2, k^{times} \times (k^{space})^2\}$, then randomly select two stages from the candidates and mutate them at each iteration step. We calculate its STEntr-Score to navigate the evolution process instead of evaluating the accuracy after mutation, if the inference cost of the mutated structure does not exceed the budget. The population will be maintained to a certain size during iterations, by discarding the worst candidate of the smallest STEntr-Score. After all iterations, the target network is achieved with the largest STEntr-Score under the given budget (*e.g.,* FLOPs, parameters, and latency). Since the latency budget requires a forward process on GPU which will diminish the efficiency of our STEntr-Score search, we choose FLOPs as the target budget. Another reason for applying FLOPs budget is to fairly compare with X3D (Feichtenhofer, 2020) and MoViNet (Kondratyuk et al., 2021), which only report FLOPs rather than latency. Thus, we obtain the spatio-temporal entropy 3D CNNs family (E3D family) under certain FLOPs, All models are searched separately with different FLOPs bugdet (1.9G, 4.7G, and 18.4G) for a fair comparison with X3D-S/M/L as the baseline, and the detailed algorithm is described in **Appendix E**

## 4 EXPERIMENTS

Our E3D family consists of E3D-S (1.9G FLOPs), E3D-M (4.7G FLOPs), and E3D-L (18.3G FLOPs). The detailed structures of the E3D family are described in **Appendix F**. We compare our approach with other state-of-the-art methods and in-depth analysis to better understand our method. More results are presented in **Appendix H**.

| Method | Backbone | Pretrain | Resolution | GFLOPs | 1×1 V1-Val | | 2×3 V2-Val | |
|---|---|---|---|---|---|---|---|---|
| | | | | | Top1 | Top5 | Top1 | Top5 |
| TSN (Wang et al., 2016) | ResNet50 | ImageNet | $8 \times 256^2$ | 16 | 19.5 | - | - | - |
| TSM (Lin et al., 2019) | ResNet50 | ImageNet | $16 \times 256^2$ | 65 | 47.2 | 77.1 | 63.4 | 88.5 |
| TANet (Liu et al., 2021) | ResNet50 | ImageNet | $16 \times 256^2$ | 66 | 47.6 | 77.7 | 64.6 | 89.5 |
| ActionNet (Wang et al., 2021b) | ResNet50 | ImageNet | $16 \times 256^2$ | 69.5 | - | - | 64.0 | 89.3 |
| TAda (Huang et al., 2022) | ConvNeXt-T | ImageNet | $16 \times 256^2$ | 47 | - | - | 64.8 | 88.8 |
| I3D (Carreira & Zisserman, 2017) | InceptionV1 | ImageNet+K400 | $64 \times 256^2$ | 306 | 41.6 | 72.2 | - | - |
| NL I3D (Carreira & Zisserman, 2017) | InceptionV1 | ImageNet+K400 | $64 \times 256^2$ | 334 | 44.4 | 76.0 | - | - |
| S3D-G (Xie et al., 2018) | InceptionV1 | ImageNet | $64 \times 256^2$ | 71.4 | 48.2 | 78.7 | - | - |
| X3D∗ (Feichtenhofer, 2020) | X3D-S | No pretrain | $13 \times 160^2$ | 2 | 44.6 | 74.4 | 60.1 | 85.9 |
| X3D∗ (Feichtenhofer, 2020) | X3D-M | No pretrain | $16 \times 224^2$ | 4.7 | 47.3 | 76.6 | 62.2 | 87.2 |
| X3D∗ (Feichtenhofer, 2020) | X3D-L | No pretrain | $16 \times 312^2$ | 18.4 | 49.4 | 77.9 | | |
| MoViNet∗ (Kondratyuk et al., 2021) | MoViNet-A0 | No pretrain | $50 \times 172^2$ | 2.7 | 46.9 | 75.0 | 61.9 | 87.2 |
| MoViNet∗ (Kondratyuk et al., 2021) | MoViNet-A1 | No pretrain | $50 \times 172^2$ | 6 | 49.3 | 77.1 | 64.5 | 89.1 |
| E3D | E3D-S | No pretrain | $13 \times 160^2$ | 1.9 | 47.1 | 75.6 | 62.1 | 87.6 |
| E3D | E3D-M | No pretrain | $16 \times 224^2$ | 4.7 | 49.4 | 78.1 | 64.7 | 89.6 |
| E3D | E3D-L | No pretrain | $16 \times 312^2$ | 18.3 | **51.1** | **78.7** | **65.7** | **89.8** |

Table 2: Comparison with state-of-the-art methods on Sth-Sth V1 and V2 validation datasets. The models only take RGB frames as inputs. To be consistent with compared approaches, we present most results of 2D CNN-based methods with ResNet50. "MN-V2"" denotes MobileNet-V2. k×k denotes temporal clip with spatial crop evaluation. "-" indicates the results are not available for us, and ∗ denotes our reproduced models.

## 4.1 Experiment Settings

The E3D family includes a search stage without training and a training & inference stage for video recognition on a specific dataset. Detailed settings in each stage are described as follows:

**Search Settings**. Following X3D (Feichtenhofer, 2020), we also apply a MobileNet-like network basis, in which the core concept is 3D depth-wise separable convolution for efficiency. The initial structure is composed of 5 stages with small and narrow blocks to meet the reasonable budget, which is usually below 1/3 of the target FLOPs budget. The population size and total iterations of EA are set as 512 and 500000, respectively. During mutation stages from the candidates, we randomly select 3D kernels from $\{1 \times 3 \times 3, 1 \times 5 \times 5, 3 \times 3 \times 3\}$ to replace the current one; interchange the expansion ratio of bottleneck from $\{1.5, 2.0, 2.5, 3.0, 3.5, 4.0\}(bottleneck = ratio \times intput)$; scale the output channels with the ratios $\{2.0, 1.5, 1.25, 0.8, 0.6, 0.5\}$; or increases or decreases depth with 1 or 2. Note that the channel dimension of every layer is fixed within 8 to 640 with multiples of 8, which helps shrink homologous search space and accelerate search speed.

**Training & Inference**. Our experiments are conducted on three large-scale datasets, Something-Something (Sth-Sth) V1&V2 (Goyal et al., 2017), and Kinetics400 (Kay et al., 2017). All models are trained by using Stochastic Gradient Descent (SGD). The cosine learning rate schedule (Loshchilov & Hutter, 2016) is employed, and total epochs are set to 100 for Sth-Sth V1&V2 datasets, and 150 for Kinetics400 dataset, with synchronized Batch-Norm instead of common Batch-Norm. Random scaling, cropping, and flipping are applied as data augmentation on all datasets. To be comparable with previous work and evaluate accuracy and complexity trade-offs, we apply two testing strategies: 1) *K-Center*: temporally, uniformly sampling of K clips (*e.g.,* K=10) from a video and taking a center crop. 2) *K-LeftCenterRight*: also uniformly sampling K clips temporally, but taking multiple crops to cover the longer spatial axis, as an approximation of fully-convolutional testing. For all methods, we follow prior studies by reporting Top1 and Top5 recognition accuracy, and FLOPs to indicate the model complexity. More experiment setting details can be seen in **Appendix G**.

## 4.2 Main Results

**Sth-Sth V1&V2**. Tabel 2 shows the comparison between E3D family and state-of-the-art methods. It can be seen that our proposed E3D family achieves competitive performance with more efficient FLOPs-level, which indicates that the E3D models can recognize actions effectively and efficiently. 1) Compared to 2D CNN-based methods, E3D outperforms most previous approaches on the same FLOPs-level. Even compared to many methods with similar performance, our model requires much lower computational costs. Note that our E3D family does not need to be pretrained on other datasets, and the performance of these 2D CNN-based methods is based on ResNet50 or a

| Method | Backbone | Pretrain | Frame | #Param. | GFLOPs × Views | Val Top1 | Val Top5 |
|---|---|---|---|---|---|---|---|
| TSN (Wang et al., 2016) | ResNet50 | ImageNet | 25 | 24.3M | 80×1×10 | 72.5 | 90.2 |
| TSM (Lin et al., 2019) | ResNet50 | ImageNet | 16 | 24.3M | 65×3×10 | 74.7 | 91.4 |
| TEA (Li et al., 2020) | ResNet50 | ImageNet | 16 | - | 70×3×10 | 76.1 | 92.5 |
| TANet (Liu et al., 2021) | ResNet50 | ImageNet | 16 | 26M | 86×3×10 | 76.9 | 92.9 |
| TDN (Wang et al., 2021a) | ResNet50 | ImageNet | 16+64 | - | 72×3×10 | 77.5 | 93.2 |
| R(2+1)D (Tran et al., 2018) | ResNet50 | ImageNet | 16 | 63.6M | 67×3×10 | 73.7 | 91.6 |
| SlowOnly (Feichtenhofer et al., 2019) | ResNet50 | ImageNet | 8 | - | 42×3×10 | 74.8 | 91.6 |
| SlowFast (Feichtenhofer et al., 2019) | ResNet50 | ImageNet | 8+32 | 34.4M | 65.7×3×10 | 77.0 | 92.6 |
| I3D (Carreira & Zisserman, 2017) | InceptionV1 | ImageNet | 64 | 12M | 108×N/A | 72.1 | 90.3 |
| Two-Stream I3D (Carreira & Zisserman, 2017) | InceptionV1 | ImageNet | 64 | 25M | 216×N/A | 75.7 | 92.0 |
| S3D-G (Xie et al., 2018) | InceptionV1 | ImageNet | 64 | - | 71.4×3×10 | 74.7 | 93.4 |
| X3D (Feichtenhofer, 2020) | X3D-M | No pretrain | 16 | 3.8M | 6.2×3×10 | 76.0 | 92.3 |
| X3D (Feichtenhofer, 2020) | X3D-L | No pretrain | 16 | 3.8M | 24.8×3×10 | 77.5 | 92.9 |
| MoViNet (Kondratyuk et al., 2021) | MoViNet-A2 | No pretrain | 50 | 4.6M | 10.3×1×1 | 75.0 | 92.3 |
| TimeSformer-S (Bertasius et al., 2021) | ViT-B | ImageNet | 8 | 121.4M | 590×3×10 | 78.0 | 93.7 |
| Swin (Liu et al., 2022) | Swin-T | ImageNet | 32 | 28.2M | 88×3×4 | 78.8 | 93.6 |
| E3D | E3D-M | No pretrain | 16 | 3.4M | 4.7×3×10 | 76.4 | 92.5 |
| E3D | E3D-L | No pretrain | 16 | 5.8M | 18.3×3×10 | 77.6 | 92.9 |

Table 3: Comparison with state-of-the-art methods on the validation set of Kinetics400. We report the inference cost with a single "view" (temporal clip with spatial crop) × the number of such views used (GFLOPs×views). "N/A" and "-" indicate the numbers are not available for us.

stronger backbone that is not suitable for low-level computation. 2) The E3D family also achieves higher performance compared to 3D CNN-based methods, which indicates that the architecture of E3D can handle the discrepancy of visual information in spatial and temporal dimensions Compared to the NAS-based method (Kondratyuk et al., 2021), our proposed E3D can still achieve a remarkable result which thus verifies the effectiveness of the STEntr-Score for searching the architecture.

**Kinetics400**. Table 3 shows that E3D achieves state-of-the-art performance compared to most 2D and 3D methods, but uses much less computational resources. 1) Most methods apply ImageNet pretrained backbones on the Kinetics400 dataset. However, our model can still achieve excellent results without using pretrained models, which indicates that our searched architecture by STEntr-Score can effectively learn spatio-temporal information. 2) E3D outperforms other 3D CNN-based models (Carreira & Zisserman, 2017; Xie et al., 2018; Feichtenhofer, 2020) which only employ 3×3×3 kernel. It means that kernel selection is important for action recognition, and STEntr-Score can benefit 3D CNN architecture design. 3) Even though the performance of Transformer-based models (Bertasius et al., 2021; Neimark et al., 2021; Liu et al., 2022) is competitive, our model still provides remarkable results by using much lower computational resources (FLOPs) and parameters, which means our model is more suitable in efficient scenarios.

## 4.3 CORRELATION STUDY

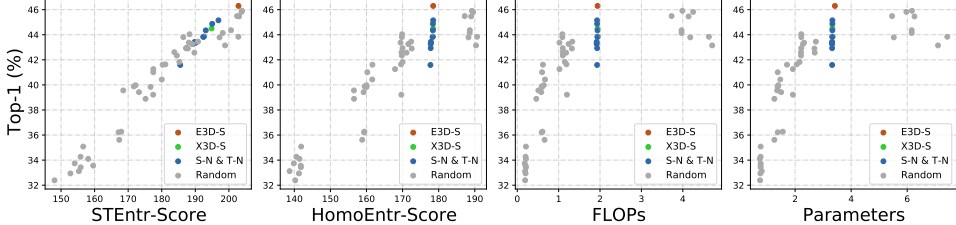

Figure 3: Correlations between Top1 and STEntr-Score, HomoEntr-Score, FLOPs, and Parameters. Points represent different sampled models, which have different channel numbers and layer configurations.

To verify the importance of STEntr-Score in the design of video understanding models, we randomly construct 60 different models (0.2 to 5 GFLOPs) with different channel dimensions and layer numbers to investigate the correlations between STEntr-Score, HomoEntr-Score, FLOPs and parameters. For a fair comparison, all networks are trained on the Sth-Sth V1 dataset with batch size of 256 and 50 epochs. We also provide the performance of E3D-S and X3D-S under the same training

setting. According to results in Figure 3, we can observe that: (1) The proposed STEntr-Score is more positively correlated with Top1 accuracy than other metrics, which proves the effectiveness of our proposed STEntr-Score in evaluating network architecture. (2) Although HomoEntr-Score is discriminative on different FLOPs levels, the ability to capture the discrepancy of the visual information in the spatial and temporal domain is not as good as STEntr-Score on the same FLOPs level. (3) Benefiting from STEntr-Score, EA can help us obtain 3D CNN architectures with higher expressiveness as measured by STEntr-Score on the same FLOPs or parameters level.

## 4.4 DISCUSSION

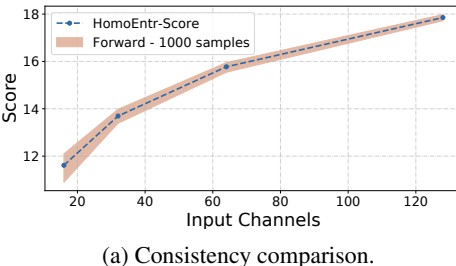
(a) Consistency comparison.

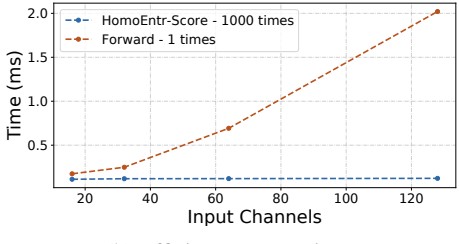
(b) Efficiency comparison.

Figure 4: Comparisons between HomoEntr-Score and "Forward" calculations. "Forward" represents using the forward inference. The calculations are conducted on an AMD Ryzen 5 5600X 6-core CPU.

**Comparison with forward inference**. For a fair comparison with the realization of forward inference in (Sun et al., 2022b), we use HomoEntr-Score and conduct a simulation in a three-layer 3D network. The shape of the input feature is $5 \times 5 \times 5$, kernel sizes are set to $1 \times 1 \times 1$, $3 \times 3 \times 3$ and $1 \times 1 \times 1$ with a stride of 1, and channels are all set to $C_{in} \in \{16, 32, 64, 128\}$. The entropy of each network is calculated $10^3$ times with either forward inference via Eq. (1) or direct computation of HomoEntr-Score. When performing the forward inference of the network, convolution blocks are re-initialized based on a Gaussian distribution during each iteration. The filled "Forward" range in Figure 4a demonstrates there exists variance between different random samples, which also emphasizes the stability of the analytic formulation. In Figure 4b, regardless of how channels change, the speed of $10^3$ times formulaic calculation of value remains constant, while the speed reduces almost linearly when performing forward inference. More comparison analysis of training-free scores is included in **Appendix D**

**Searching cost comparison**. Since we apply analytic formulation rather than inference, the calculation of our STEntr-Score has lower hardware requirements, which means that CPU resources can meet it instead of GPU or TPU. From Table 13, our method only takes three hours of searching time with a desktop CPU, while MoViNet consumes 24 hours with 64 commercial TPUs. Extremely low time and power consumption demonstrate the searching efficiency of our analytic entropy formulation.

| Method | Search Devices | Search Time | Power Consumption | GFLOPs | TOP-1 |
|---|---|---|---|---|---|
| MoViNet-A1 | TPUs § | 24h | 691.2kWh | 6 | 49.3 |
| E3D-M | CPU † | 3h | 0.195kWh | 4.7 | 49.4 |

Table 4: Searching cost comparison on the Sth-Sth V1 dataset. §: 64 Google TPUv3, Power 450W per TPUv3; †: 1 AMD Ryzen 5 5600X 6-Core CPU, Power 65W;

## 5 CONCLUSION

In this paper, we propose to automatically design efficient 3D CNN architectures via an entropy-based training-free neural architecture search approach, to address the problem of efficient action recognition. In particular, we first formulate the 3D CNN architecture as an information system and propose the STEntr-Score to measure the expressiveness of the system. Then we obtain the E3D family by an evolutionary algorithm, with the help of STEntr-Score. Extensive results show that our searched E3D family achieves higher accuracy and better efficiency compared to many state-of-the-art action recognition models, within three desktop CPU hours searching.

ACKNOWLEDGMENTS

This research was supported by Alibaba Group through Alibaba Research Intern Program, and ARC DECRA Fellowship DE230101591 to D. Gong.

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

APPENDIX

In the appendix, we provide a detailed description of notations in this paper (Appendix A), detailed proof of equations (Appendix B), a comparison between different training-free scores on the ImageNet dataset (Appendix C), a discussion of simple network space in the entropy mechanism (Appendix D), STEntr-Score for maximizing expressiveness (Appendix E), E3D family structure details (Appendix F), experimental setting details (Appendix G), additional result analysis (Appendix H), and future work discussion (Appendix I).

## A MEANING OF NOTATIONS

| Notation | Size | Meaning |
|---|---|---|
| $\mathcal{H}(x)$ | - | The function of computing the entropy of the given $x$ |
| $P(x)$ | - | The distribution of the given $x$ |
| $\mu$ | Constant | The value of expectation |
| $\mathbb{E}$ | - | The function of computing expectation |
| $\sigma$ | Constant | The value of variance |
| $\mathbb{D}$ | - | The function of computing variance |
| $K_t$ | Constant | Temporal dimension size of 3D CNN kernel |
| $K_h$ | Constant | Height dimension size of 3D CNN kernel |
| $K_w$ | Constant | Width dimension size of 3D CNN kernel |
| $C$ | Constant | Channel dimension size |
| $\boldsymbol{K}$ | $K_t \times K_h \times K_w$ | A 3D CNN kernel size |
| $\boldsymbol{W}$ | $C \times \boldsymbol{K}$ | The weight matrix of the CNN layer |
| $\boldsymbol{S}$ | $T \times H \times W$ | The input feature map size of a given depth (time $\times$ height $\times$ width) |
| $\mathcal{D}_{cosine}$ | - | The cosine similarity distance function |
| $\hat{\mathcal{D}}$ | - | The expanded diversity of cosine similarity distance function |

Table 5: The meaning of all notations appeared in this paper.

## B PROOF OF SPATIO-TEMPORAL ENTROPY SCORE

### B.1 DERIVATION PROCESS OF DIFFERENTIAL ENTROPY

Suppose $x$ is sampled from Gaussian distribution $\mathcal{N}(\mu, \sigma^2)$, and we know about the probability density function of $x$:

$$p(x) = \frac{1}{\sqrt{2\pi}\sigma} exp[-\frac{(x-\mu)^2}{2\sigma^2}] \,. \tag{9}$$

We can then derive the differential entropy with $\int_{-\infty}^{+\infty} e^{-x^2} dx = \sqrt{\pi}$ as:

$$
\begin{aligned}
\mathcal{H}(x) &= \int_{-\infty}^{+\infty} -log(p(x)p(x)dx \\
&= -\int_{-\infty}^{+\infty} \frac{1}{\sqrt{2\pi}\sigma} exp[-\frac{(x-\mu)^2}{2\sigma^2}] log \frac{1}{\sqrt{2\pi}\sigma} exp[-\frac{(x-\mu)^2}{2\sigma^2}] dx \\
&= \frac{log(\sqrt{2\pi}\sigma)}{\sqrt{\pi}} \int_{-\infty}^{+\infty} e^{-y^2} dy + \frac{1}{\sqrt{\pi}} \int_{-\infty}^{+\infty} e^{-y^2} y^2 dy \\
&= log(\sqrt{2\pi}\sigma) + \frac{1}{\sqrt{\pi}} \times [-\frac{1}{2}(0 - \int_{-\infty}^{+\infty} e^{-y^2} dy)] \\
&= \frac{1}{2}log(2\pi) + log(\sigma) + \frac{1}{2} \propto log(\sigma^2) \,.
\end{aligned}
\tag{10}
$$

## B.2  EXPECTATION AND VARIANCE OF FEATURE MAP

According to **Theorem 2** and **Theorem 3**, we can compute the expectation of $l^{th}$ layer feature map element $\boldsymbol{x}_i^l$, as:

$$\mathbb{E}(\boldsymbol{x}_i^l) = \mathbb{E}(\sum_{t=1}^{K_t^l}\sum_{h=1}^{K_h^l}\sum_{w=1}^{K_w^l}\sum_{c=1}^{C^{l-1}} \boldsymbol{x}_{cthw}^{l-1}\boldsymbol{W}_{cthw}^l) = \sum_{t=1}^{K_t^l}\sum_{h=1}^{K_h^l}\sum_{w=1}^{K_w^l}\sum_{c=1}^{C^{l-1}} \left[\mathbb{E}(\boldsymbol{x}_{cthw}^{l-1})\mathbb{E}(\boldsymbol{W}_{cthw}^l)\right], \qquad (11)$$

Given two independent random variables $v_1$ and $v_2$, based on $\mathbb{D}(v) = \mathbb{E}(v^2) - \mathbb{E}(v)^2$ and **Theorem 2**, we can then calculate the variance of the product of these variables as:

$$\begin{aligned}\mathbb{D}^2(v_1 v_2) &= \mathbb{E}(v_1^2 v_2^2) - \mathbb{E}(v_1 v_2)^2 = \mathbb{E}(v_1^2)\mathbb{E}(v_2^2) - \mathbb{E}(v_1)^2\mathbb{E}(v_2)^2 \\ &= [\mathbb{D}(v_1) + \mathbb{E}(v_1)^2][\mathbb{D}(v_2) + \mathbb{E}(v_2)^2] - \mathbb{E}(v_1)^2\mathbb{E}(v_2)^2 \\ &= \mathbb{D}^2(v_1)\mathbb{D}^2(v_2) + \mathbb{D}^2(v_2)[\mathbb{E}(v_1)]^2 + \mathbb{D}^2(v_1)[\mathbb{E}(v_2)]^2 , \end{aligned} \qquad (12)$$

We can then derive the variance of $\boldsymbol{x}_i^l$, based on **Theorem 2** and **Theorem 3**, as:

$$\begin{aligned}\mathbb{D}^2(\boldsymbol{x}_i^l) &= \mathbb{D}^2(\sum_{t=1}^{K_t^l}\sum_{h=1}^{K_h^l}\sum_{w=1}^{K_w^l}\sum_{c=1}^{C^{l-1}} \boldsymbol{x}_{cthw}^{l-1}\boldsymbol{W}_{cthw}^l) \\ &= \sum_{t=1}^{K_t^l}\sum_{h=1}^{K_h^l}\sum_{w=1}^{K_w^l}\sum_{c=1}^{C^{l-1}} \mathbb{D}^2(\boldsymbol{x}_{cthw}^{l-1}\boldsymbol{W}_{cthw}^l) \\ &= \sum_{t=1}^{K_t^l}\sum_{h=1}^{K_h^l}\sum_{w=1}^{K_w^l}\sum_{c=1}^{C^{l-1}} \Big\{\mathbb{D}^2(\boldsymbol{x}_{cthw}^{l-1})\mathbb{D}^2(\boldsymbol{W}_{cthw}^l) \\ &\quad + \mathbb{D}^2(\boldsymbol{x}_{cthw}^{l-1})\Big[\mathbb{E}(\boldsymbol{W}_{cthw}^l)\Big]^2 + \mathbb{D}^2(\boldsymbol{W}_{cthw}^l)\Big[\mathbb{E}(\boldsymbol{x}_{cthw}^{l-1})\Big]^2\Big\}, \end{aligned} \qquad (13)$$

## B.3  PROOF OF 3D CNNs ENTROPY

As the input $\boldsymbol{x}^0$ is initialized from a standard Gaussian distribution $\mathcal{N}(0,1)$, and all parameters initialized from Gaussian Distribution $\mathcal{N}(0, \sigma_w^2)$, we can formulate Eq. (11) and Eq. (4) as:

$$\mathbb{E}(\boldsymbol{x}_i^1) = 0, \quad \mathbb{D}^2(\boldsymbol{x}_i^1) = \sum_{t=1}^{K_t^1}\sum_{h=1}^{K_h^1}\sum_{w=1}^{K_w^1}\sum_{c=1}^{C^0} \left[\mathbb{D}^2(\boldsymbol{W}_{chw}^1)\right], \qquad (14)$$

Subsequently, the expectation $\mathbb{E}(\boldsymbol{x}_i^L)$ and variance $\mathbb{D}^2(x^L)_i$ of the last layer can be derived as:

$$\mathbb{E}(\boldsymbol{x}_i^L) = 0, \quad \mathbb{D}^2(\boldsymbol{x}_i^L) = \sum_{t=1}^{K_t^L}\sum_{h=1}^{K_h^L}\sum_{w=1}^{K_w^L}\sum_{c=1}^{C^{L-1}} \left[\mathbb{D}^2(\boldsymbol{W}_{chw}^L)\right], \qquad (15)$$

Therefore, the variance can be computed by propagating the variances from previous layers as:

$$\mathbb{D}^2(\boldsymbol{x}_i^L) = \prod_{l=1}^{L} K_t^l K_h^l K_w^l C^{l-1}\mathbb{D}^2(\boldsymbol{W}_{chw}^l), \qquad (16)$$

According to Eq. (1), the upper bound entropy is proportional to the variance of last feature map. Then we can derive Eq. (1) as:

$$\mathcal{H}(F) \propto \sum_{l=1}^{L} log(K_t^l K_h^l K_w^l C^{l-1}\mathbb{D}^2(\boldsymbol{W}_{chw}^l)) , \qquad (17)$$

## C  DISCUSSION OF SIMPLE NETWORK SPACE

The bias of a convolutional layer is zero, and the activation function in the network is omitted in the search for simplification, following the work of ZenNAS (Lin et al., 2021) and MAE-DET (Sun et al., 2022b), which has been shown to have no influence on the expressiveness of the network. The training of CNN models has been well studied, and some components can be integrated to boost performance. We deliberately avoid using these components to keep our design simple and universal. Nevertheless, these auxiliary components can easily be plugged into the architecture without any special modifications. Moreover, we provide a discussion of auxiliary components with the entropy calculation, which is listed below.

**Batch Normalization (BN)**. BN is a widely used method to re-center and re-scale the features to make the network converge faster and more stable. BN normalizes entropies adaptively to the network width (which can be related to output variance). When BN is used, networks of different widths will have the same entropy value. Hence, BN has to be removed when calculating entropy.

**Activation Function**. Activation functions increase the non-linearity of training, which has different effects on entropy. For example, ReLUs, half the variance of the output, decrease the entropy with a constant factor in each layer, having a less positive effect on entropy. Meanwhile, if we formulate each kind of activation for our system, it introduces redundancy and becomes complicated, so we give them a uniform form to omit them in search of concise expressiveness calculation.

**Residual Link**. If the input and all parameters are initialized from standard Gaussian distribution, the variances with or without residual links are less than 2% different in entropy score, which means it affects the entropy value slightly. Meanwhile, the residual link has a significant impact on convergence in training.

**Squeeze-and-Excitation Module (SE)**. SE modules are used to adaptively recalibrate channel-wise feature responses by explicitly modeling interdependency between channels. When the input is initialized from a Gaussian distribution, the output after global pooling in SE block is equal to 0 and the final output becomes 0.5, which will lose the ability to model interdependency between channels.

## D  COMPARISON ON TRAINING-FREE SCORES

### D.1  COMPARISON ON THE IMAGENET-1K DATASET.

| Training-free method | FLOPs | Search Devices | Design Cost (hours) | Top-1 |
|---|---|---|---|---|
| ResNet-50 | 4.1G | - | - | 78.0 |
| Zen-score (Lin et al., 2021) | 4.4G | GPU‡ | 24 | 78.9 |
| MAE-DET score (Sun et al., 2022b) | 4.4G | GPU‡ | 14 | 79.1 |
| HomoEntr-Score (w/o $K^t$) | 4.3G | CPU† | 3 | 79.0 |

Table 6: Comparison of different training-free methods on ImageNet-1K dataset. ‡: Nvidia Tesla V100 16G GPU, †: AMD Ryzen 5 5600X 6-Core CPU.

We conduct comparison experiments using 2D CNNs on ImageNet with the same evolutionary strategies (ResNet design space), as shown in Table 6. Compared with the result of ResNet-50, the model searched by HomoEntr-Score improves 1.0% of accuracy, which indicates that the entropy-based analytic formulation can also measure the information capacity of 2D CNNs. Compared with Zen-score and MAE-DET, the performance of our proposed formulaic metric can also achieve comparable performance. It means that HomoEntr-Score can work well for modeling the information capacity of 2D CNNs, since there is no (obvious) discrepancy in the information of the two directions in 2D images statistically.

## D.2 COMPARISON ON THE STH-STH V1 DATASET.

Since there is no existing code available for training-free NAS methods for 3D CNNs, we then refine their implementations for the video recognition task. The results are shown in Table 7.

| Training-free method | GFLOPs | Search Devices | Design Cost (hours) | Top-1 | Top-5 |
|---|---|---|---|---|---|
| X3D-S (Feichtenhofer, 2020) | 2G | - | - | 44.6 | 74.4 |
| Zen-score (Lin et al., 2021) | 1.9G | GPU | 26 | 45.5 | 74.6 |
| MAE-DET score (Sun et al., 2022b) | 1.9G | GPU | 15 | 45.8 | 74.7 |
| E3D-S | 1.9G | CPU | 3 | 47.1 | 75.6 |

Table 7: Comparison of different training-free methods on the Sth-Sth V1 dataset. ‡: Nvidia Tesla V100 16G GPU, †: AMD Ryzen 5 5600X 6-Core CPU.

According to the results in Table 7, the performances of other training-free NAS methods are better than X3D-S, but the performance of our searched model is higher. It indicates directly applying training-free NAS methods can be effective in the video recognition task, but it still needs spatio-temporal refinement on video understanding tasks, which our work mainly focuses on.

## E DETAILED SEARCHING ALGORITHM AND SETTINGS

To obtain highly expressive 3D CNNs of maximized entropy, we use a customized Evolutionary Algorithm. The step-by-step description of EA is given in Algorithm 1, as the architecture generator. We only apply the STEntr-Score to guide the evolution process, not accuracy, which therefore does not need training on the dataset. We choose EA due to its simplicity, and it is possible to choose other methods, such as reinforcement learning or even greedy selection. According to our kernel selection observations, we define the 3D kernel size search space within each layer, $1 \times (k^{space})^2$, $k^{times} \times (k^{space})^2$, to be chosen as one of the following: $\{1\times3\times3, 1\times5\times5, 3\times3\times3\}$. These choices enable a layer to focus on and aggregate different dimensional representations efficiently, expanding the network's receptive field in the most pertinent directions, while reducing FLOPs along other dimensions (Kondratyuk et al., 2021).

### E.1 INITIAL ARCHITECTURE

| Stage | Kernels | Channels | Layers | $T \times H \times W$ |
|---|---|---|---|---|
| data | stride 6, $1^2$ | 3 | 1 | $13 \times 160 \times 160$ |
| $conv_1$ | $1 \times 3^2$, 24 | 24 | 1 | $13 \times 80 \times 80$ |
| $stage_2$ | $[1\times1^2, 3\times3^2, 1\times1^2]$ | [48, 48, 24] | 1 | $13 \times 40 \times 40$ |
| $stage_3$ | $[1\times1^2, 3\times3^2, 1\times1^2]$ | [96, 96, 48] | 1 | $13 \times 20 \times 20$ |
| $stage_4$ | $[1\times1^2, 3\times3^2, 1\times1^2]$ | [192, 192, 96] | 1 | $13 \times 10 \times 10$ |
| $stage_5$ | $[1\times1^2, 3\times3^2, 1\times1^2]$ | [192, 192, 96] | 1 | $13 \times 10 \times 10$ |
| $stage_6$ | $[1\times1^2, 3\times3^2, 1\times1^2]$ | [384, 384, 192] | 1 | $13 \times 5 \times 5$ |
| $conv_7$ | $1 \times 1^2$ | 512 | 1 | $13 \times 5 \times 5$ |
| $pool_8$ | $13 \times 5 \times 5$ | 512 | 1 | $1 \times 1 \times 1$ |
| $conv_{9/10}$ | $1 \times 1^2, 1 \times 1^2$ | [2048, #classes] | 1 | $1 \times 1 \times 1$ |

Table 8: E3D-S initial searching. "$stage_x$" is a super structure which contains "layers"-layer 3D inverted bottleneck block. Channels means the output channels of the corresponding convolution.

Firstly, we set up the initial architecture in a MobileNet-styled network, as shown in Table 8, which consists of five stages with only one layer that can be easily evolved during the algorithm. The

initial architecture is inspired by the structure of X3D-S (Feichtenhofer, 2020) because inheriting good prior design can reduce the uncertainty of search space. Then, based on the initial architecture, applied EA helps us mutate channel dimension, kernel selection, bottleneck expansion ratio, and layer arrangement by randomly selecting the stage. Note that the channel dimension in $conv_1$ and $conv_7$ also participate in the mutation process.

---

**Algorithm 1** Maximum Entropy Evolutionary Algorithm

---

**Require:** Search space $\mathcal{S}$. Inference budget $B$, maximal depth $L$, total number of iterations $M$, evolutionary population size $N$, initial structure $F_0$.
**Ensure:** Designed E3D backbone $F^*$.
 1: Initialize population $\mathcal{P} = \{F_0\}$.
 2: **for** $m = 1, 2, \cdots, M$ **do**
 3:     Randomly select $F_m \in \mathcal{P}$ and select two stages $stage_k \in F_m$.
 4:     **for** $j = 1, 2$ **do**
 5:         **Switch** Randomly select one target of {**Kernel** size, **Output** channels, **Bottleneck** channels, **Layers**} from $stage_{kj}$ **do**
 6:         **Case kernel:** Mutate kernel from 3D kernel search space.
 7:         **Case Output:** Mutate output channels with multiplier space.
 8:         **Case Bottleneck:** Mutate bottleneck channels with expansion ratio space.
 9:         **Case Layers:** Mutate block layers with addend from $\{-2, -1, 1, 2\}$.
10:     **end for**
11:     Get mutated network $\hat{F}_m$ with two mutated stages $stage_{kj}$.
12:     **if** $\hat{F}_m$ is within inference budget $B$ and has no more than $L$ layers **then**
13:         Get STEntr-Score of $\hat{F}_m$ and append $\hat{F}_m$ to $\mathcal{P}$.
14:     **end if**
15:     Remove networks of the smallest STEntr-Score if the size of $\mathcal{P}$ exceeds $B$.
16: **end for**
17: Return $F^*$, the network of the highest STEntr-Score in $\mathcal{P}$.

---

### E.2 EVOLUTIONARY ALGORITHM

In Algorithm 1, we randomly initialize a population of candidates from the initial structure, under a computational budget. The population size and total iterations of EA are set to 512 and 500000, respectively. At each iteration step $m$, we randomly select two stages from the candidates and mutate them. Next, we will randomly select a mutation strategy from 4 strategies for each stage. Specific mutation strategies for our E3D family are described as follows. We randomly select 3D kernels from $\{1\times3\times3, 1\times5\times5, 3\times3\times3\}$ to replace the current one; interchange the expansion ratio of bottleneck from $\{1.5, 2.0, 2.5, 3.0, 3.5, 4.0\}(bottleneck = ratio \times intput)$; scale the output channels with the ratios $\{2.0, 1.5, 1.25, 0.8, 0.6, 0.5\}$; or increases or decreases depth with 1 or 2. Note that the channel dimension of every layer is fixed within from 8 to 640 with multiples of 8, which will help shrink homologous search space and accelerate the search speed. The mutated structure $\hat{F}_m$ is appended to the population if its inference cost does not exceed the budget. Finally, we maintain the population size by removing networks with the smallest STEntr-Score. After $M$ iterations, the target network with the largest STEntr-Score is obtained, namely E3D.

## F E3D FAMILY ARCHITECTURE DETAILS

Table 9 shows three instantiations of E3D with varying complexity, including E3D-S (1.9G FLOPs), E3D-M (4.7G FLOPs), and E3D-L (18.3G FLOPs). All models are searched separately with different FLOPs budget (1.9G, 4.7G, and 18.4G) for a fair comparison with X3D-S/M/L as the baseline. Meanwhile, SE block and ReLU activation function will be added into these architectures for training. For both training and inference, the input size remains the same: 160 for E3D-S, 224 for E3D-M, and 312 for E3D-L. All channel dimensions and layer arrangements are searched by evolutionary algorithm under different given budgets.

| Stage | E3D-S | | E3D-M | | E3D-L | |
|---|---|---|---|---|---|---|
| | filters | output size | filters | output size | filters | output size |
| data | stride 6, $1^2$ | $13 \times 160 \times 160$ | stride 5, $1^2$ | $16 \times 224 \times 224$ | stride 5, $1^2$ | $16 \times 312 \times 312$ |
| $conv_1$ | $1 \times 3^2, 24$ | $13 \times 80 \times 80$ | $1 \times 3^2, 24$ | $16 \times 112 \times 112$ | $1 \times 3^2, 24$ | $16 \times 156 \times 156$ |
| $stage_2$ | $\begin{bmatrix} 1 \times 1^2, 32 \\ 1 \times 5^2, 32 \\ 1 \times 1^2, 24 \end{bmatrix} \times 3$ | $13 \times 40 \times 40$ | $\begin{bmatrix} 1 \times 1^2, 32 \\ 1 \times 5^2, 32 \\ 1 \times 1^2, 24 \end{bmatrix} \times 3$ | $16 \times 56 \times 56$ | $\begin{bmatrix} 1 \times 1^2, 32 \\ 1 \times 5^2, 32 \\ 1 \times 1^2, 24 \end{bmatrix} \times 3$ | $16 \times 78 \times 78$ |
| $stage_3$ | $\begin{bmatrix} 1 \times 1^2, 96 \\ 3 \times 3^2, 96 \\ 1 \times 1^2, 48 \end{bmatrix} \times 6$ | $13 \times 20 \times 20$ | $\begin{bmatrix} 1 \times 1^2, 96 \\ 3 \times 3^2, 96 \\ 1 \times 1^2, 64 \end{bmatrix} \times 6$ | $16 \times 28 \times 28$ | $\begin{bmatrix} 1 \times 1^2, 120 \\ 3 \times 3^2, 120 \\ 1 \times 1^2, 48 \end{bmatrix} \times 13$ | $16 \times 39 \times 39$ |
| $stage_4$ | $\begin{bmatrix} 1 \times 1^2, 176 \\ 3 \times 3^2, 176 \\ 1 \times 1^2, 120 \end{bmatrix} \times 6$ | $13 \times 10 \times 10$ | $\begin{bmatrix} 1 \times 1^2, 176 \\ 3 \times 3^2, 176 \\ 1 \times 1^2, 120 \end{bmatrix} \times 6$ | $16 \times 14 \times 14$ | $\begin{bmatrix} 1 \times 1^2, 176 \\ 3 \times 3^2, 176 \\ 1 \times 1^2, 120 \end{bmatrix} \times 13$ | $16 \times 20 \times 20$ |
| $stage_5$ | $\begin{bmatrix} 1 \times 1^2, 176 \\ 3 \times 3^2, 176 \\ 1 \times 1^2, 120 \end{bmatrix} \times 6$ | $13 \times 10 \times 10$ | $\begin{bmatrix} 1 \times 1^2, 176 \\ 3 \times 3^2, 176 \\ 1 \times 1^2, 120 \end{bmatrix} \times 6$ | $16 \times 14 \times 14$ | $\begin{bmatrix} 1 \times 1^2, 176 \\ 3 \times 3^2, 176 \\ 1 \times 1^2, 120 \end{bmatrix} \times 13$ | $16 \times 20 \times 20$ |
| $stage_6$ | $\begin{bmatrix} 1 \times 1^2, 384 \\ 3 \times 3^2, 384 \\ 1 \times 1^2, 256 \end{bmatrix} \times 6$ | $13 \times 5 \times 5$ | $\begin{bmatrix} 1 \times 1^2, 464 \\ 3 \times 3^2, 464 \\ 1 \times 1^2, 184 \end{bmatrix} \times 6$ | $16 \times 7 \times 7$ | $\begin{bmatrix} 1 \times 1^2, 480 \\ 3 \times 3^2, 480 \\ 1 \times 1^2, 192 \end{bmatrix} \times 13$ | $16 \times 10 \times 10$ |
| $conv_7$ $pool_8$ $conv_{9/10}$ | $1 \times 1^2,$ $13 \times 5 \times 5$ [2048, #classes] | $13 \times 5 \times 5$ $1 \times 1 \times 1$ $1 \times 1 \times 1$ | $1 \times 1^2, 464$ $16 \times 7 \times 7$ [2048, #classes] | $16 \times 7 \times 7$ $1 \times 1 \times 1$ $1 \times 1 \times 1$ | $1 \times 1^2, 480$ $16 \times 10 \times 10$ [2048, #classes] | $16 \times 10 \times 10$ $1 \times 1 \times 1$ $1 \times 1 \times 1$ |

Table 9: Three instantiations of E3D with varying complexity. E3D-S with 1.9G FLOPs, E3D-M with 4.7G FLOPs, and E3D-L with 18.4G FLOPs. The size of output is $T \times H \times W$.

# G  EXPERIMENT SETTING DETAILS

## G.1  DATASETS

Our experiments are conducted on three large-scale datasets: Something-Something (Sth-Sth) V1&V2 (Goyal et al., 2017), and Kinetics400 (Kay et al., 2017). More dataset details can be seen in the supplementary materials. 1) The Sth-Sth datasets are more focused on fine-grained and motion-dominated actions, which contain pre-defined basic actions involving different interacting objects. Sth-Sth V1 comprises 86k video clips in the training set and 12k video clips in the validation set. Sth-Sth V2 is an updated version of Sth-Sth V1, which contains 169k video clips in the training set and 25k video clips in the validation set. They both have 174 action categories. 2) The Kinetics dataset contains activities in daily life and some categories are highly correlated with interacting objects or scene context. Kinetics400 contains over 200k training videos and 20k validation videos divided into 400 categories, covering a wide range of human activities.

## G.2  IMPLEMENTATION DETAILS

Detailed implementation settings of **training & inference stage** on Sth-Sth V1&V2 and Kinetics400 datasets are listed in Table 10. All experiments are performed on 8×Nvidia Tesla A100 GPUs.

| Hyperparameter | Sth-Sth V1&V2 | Kinetics400 |
|---|---|---|
| Epoch | 128 | 256 |
| Batch Size per GPU | 32 | 16 |
| Optimizer | SGD | SGD |
| Learning Rate | 0.8 | 0.4 |
| Learning Rate Policy | cosine | cosine |
| Momentum | 0.9 | 0.9 |
| Weight Decay | $5e^{-5}$ | $5e^{-5}$ |
| Warm-up Epoch | 10 | 15 |
| Synchronized Batch Normalization | True | True |
| Training from scratch | True | True |

Table 10: List of hyperparameters used on Sth-Sth V1&V2 and Kinetics400 datasets.

# H ADDITIONAL RESULTS

## H.1 ACCURACY VS. COMPLEXITY

Figure 5 shows the trade-off between accuracy and complexity (FLOPs). Compared to 2D CNN-based methods, E3D requires much lower computational resources. Although the performance of our method is similar to Tada-R50, the FLOPs of Tada-R50 are 4.7 times more than E3D-L. Compared to 3D CNN-based methods, we observe that both E3D and MoViNet can achieve large improvement, which indicates that searched methods have higher efficiency in utilizing computing resources. Also, our method achieves comparable performance compared with MoViNet, which indicates that the proposed training-free STEntr-Score can effectively evaluate the expressiveness of a 3D architecture.

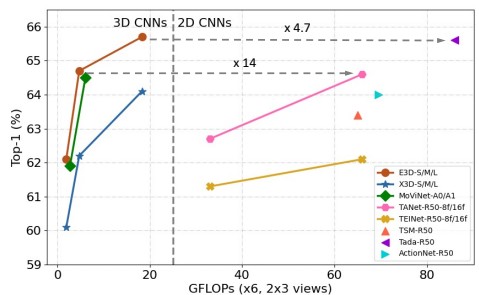

Figure 5: Accuracy/complexity trade-off on the Sth-Sth V2 dataset.

## H.2 HOMOENTR-SCORE VS. STENTR-SCORE

Table 11 reports E3D results searched by HomoEntr-Score and STEntr-Score, under the same search settings. The results show substantial improvement when using STEntr-Score instead of HomoEntr-Score, which indicates the effectiveness of STEntr-Score to handle the discrepancy of visual information in spatial and temporal dimensions. Even though without refinement factor, the performance of HomoEntr-Score searched E3D still outperforms X3D, which means the entropy-based search strategy can also measure the expressiveness of 3D CNN architectures.

| Model | Resolution | GFLOPs | Top-1 | Top-5 |
|---|---|---|---|---|
| X3D-S* (Feichtenhofer, 2020) | $13 \times 160^2$ | 2 | 44.6 | 74.4 |
| MoViNet-A0* (Kondratyuk et al., 2021) | $50 \times 172^2$ | 2.7 | 46.9 | 75.0 |
| E3D (HomoEntr-Score) | $13 \times 160^2$ | 1.9 | 45.8 | 74.8 |
| E3D (STEntr-Score) | $13 \times 160^2$ | 1.9 | 47.1 | 75.6 |

Table 11: Comparison of different entropy scores on the Sth-Sth V1 dataset. * denotes our reproduced models.

## H.3 3D KERNEL SEARCH SPACE

To analyze the impact of kernel search space, we expand the 3D kernel search space and conduct experiments, as shown in Table 12. The results indicate that larger search spaces actually benefit the performance. However, compared to the results between E3D (HomoEntr-Score) with E3D (STEntr-Score)) in Table 11, the STEntr-Score based searching can boost the performance (+1.3%) more than a large search space did (+0.2%). It also verified the effectiveness of our proposed STEntr-Score in evaluating the expressiveness of 3D CNNs.

| Kernel Search Space | FLOPs | Top-1 | STEntr-Score |
|---|---|---|---|
| $1 \times 3 \times 3$, $3 \times 3 \times 3$ | 1.9G | 46.3 | 198.55 |
| $1 \times 3 \times 3$, $1 \times 5 \times 5$, $3 \times 3 \times 3$ | 1.9G | 47.1 | 202.86 |
| $1 \times 3 \times 3$, $1 \times 5 \times 5$, $3 \times 3 \times 3$, $3 \times 1 \times 1$ | 1.9G | 47.1 | 202.74 |
| $1 \times 3 \times 3$, $1 \times 5 \times 5$, $3 \times 3 \times 3$, $5 \times 3 \times 3$ | 1.9G | 47.2 | 203.13 |

Table 12: Comparison of different 3D kernal search space on the Sth-Sth V1 dataset.

## H.4 INFERENCE TIME COMPARISON

We report the inference time comparison with some state-of-the-art methods in Table 13. All models are trained and tested on the Sth-Sth V1 dataset, and the batch size is set to 16. Compared to X3D, our E3D performs better not only on accuracy but also costs lower inference time. It indicates that the searched architecture by our proposed STEntr-Score is more effective and efficient for video understanding. Compared to MoViNet, even though Top-1 accuracies are similar, both latency and throughput of E3D are performing better. Due to MoViNet applies a causal convolutional network and contains more parameters. Compared to 2D CNN-based methods, E3D performs better on both accuracy and running time and requires much lower computational resources. Overall, we believe that our proposed E3D family is more efficient and practical for real-world applications.

| Method | Resolution | Frame | GFLOPs | #Param | Top1 | Latency (ms/video) | Throughput(video/s) |
|---|---|---|---|---|---|---|---|
| TSM (Lin et al., 2019) | 256 | 16 | 65 | 23.9M | 47.2 | 23.0 | 43.5 |
| TANet (Liu et al., 2021) | 256 | 16 | 66 | 26M | 47.6 | 14.7 | 68.0 |
| X3D-M (Feichtenhofer, 2020) | 224 | 16 | 4.7 | 3.7M | 47.3 | 13.5 | 74.1 |
| MoViNet-A1 (Kondratyuk et al., 2021) | 172 | 50 | 6 | 4.6M | 49.3 | 21.9 | 45.7 |
| E3D-M | 224 | 16 | 4.7 | 3.4M | 49.4 | 11.4 | 87.7 |

Table 13: Inference comparison using a Tesla V100 on the Sth-Sth V1 dataset.

## I  FUTURE DIRECTION

**Data-driven design**. The design of STEntr-Score search correlates with parameter initialization and kernel selection, with standard Gaussian initialization input. If we replace the Gaussian input directly with target data, the output after a convolution will be random due to the Gaussian initialized weights, as the process of STEntr-Score based searching is contained without data training. The aim of our work is therefore to provide a training-free approach to 3D CNN architecture design according to the maximum entropy principle under the given budgets. We believe that the training-free method, combined with target data without training, could be a future direction for research.

**Transformer model**. We believe that the principle of maximum entropy is theoretically applicable to transformers. However, there exist some challenges to overcome. For example, Transformer has more complex components than CNN, such as 'Q' and 'K' kernel operation and multi-head attention, which is difficult to calculate the maximum entropy. In addition, the discrepancy of visual information in spatial and temporal dimensions by Transformer still remains a challenge. Although these challenges are difficult to overcome, this would be a fascinating task for us in the future.

