# OpenReview forum: "Maximizing Spatio-Temporal Entropy of Deep 3D CNNs for Efficient Video Recognition"
_ICLR.cc/2023/Conference — ICLR 2023 poster_

### Official Review · Reviewer_Jkpo · 2022-10-25

**Confidence:** 3
**Correctness:** 2
**Technical Novelty And Significance:** 2
**Empirical Novelty And Significance:** 2
**Recommendation:** 6

**Clarity, Quality, Novelty And Reproducibility:**

The paper is well-written and the theorems are carefully proven. However, the experiments are not sufficient and the baselines seem outdated.

**Strength And Weaknesses:**

## Strength

1. The derivation of the entropy score is thorough.
1. The visualization of the searched architectures in Figure 1 reveals interesting properties of 3D CNNs.

## Weakness

1. As the search space is the most important part of a NAS method, the authors should give more details such as the lower bound / upper bound performance of the search space.
1. The search happens without considering auxiliary modules like BN, Reslink, and so on. Will the entropy score transfer well with such a gap?
1. Why is Table 2 only compared with most 2D CNN-based methods with ResNet50? Also, the baselines are outdated.


**Summary Of The Paper:**

This paper proposes a spatio-temporal entropy score to conduct training-free neural architecture search for efficient video recognition. They use a simple network search space with a carefully designed performance estimation score. The searched results achieve state-of-the-art performance on action recognition.

**Summary Of The Review:**

Baselines are most 2D CNNs-based methods and outdated. So I recommend to weak reject this paper until results on stronger baselines are provided.

---

> ### Author Response · Authors · 2022-11-16
> **Response to Reviewer Jkpo**
>
> Thanks for your comments. Please see below our response to the specific weakness.
>
> > W1: As the search space is the most important part of a NAS method, the authors should give more details such as the lower bound / upper bound performance of the search space.
>
> A1: The detailed search settings and description are provided in Section 4.1 and Appendix E.
> We also provided a search space comparison experiment in Appendix H.3.
> According to Section 4.1, the minimal model in the search space consists of channel number of 8, kernel size of 1X3X3, bottleneck expansion ratio of 1.5, and block depth of 1, and the Top1 accuracy is 12.6% on the Sth-Sth v1 datasets compared with the E3D-S 47.1%.
> Due to the unlimited depth setting of increasing 1 or 2 layers for mutation, the number of candidate networks in our search space is infinite.
> Therefore, the upper bound performance of our search space should be obtained within a given budget, as described in Algorithm 1.
> For example, the E3D-L model is the optimal model under the given 18.3GFLOPs budget.
>
> > W2: The search happens without considering auxiliary modules like BN, Reslink, and so on. Will the entropy score transfer well with such a gap?
>
> A2: The detailed discussion of auxiliary components with the entropy calculation is provided in Appendix C.
> The simple network design was widely used in previous training-free NAS methods, such as ZenNAS [1] and MAE-DET [2], which has shown that there is no influence on the expressiveness of the network.
> We deliberately avoid using these components to keep our score simple and universal when searching. Nevertheless, these auxiliary components can easily be plugged into the searched architecture during training.
>
> > W3: Why is Table 2 only compared with most 2D CNN-based methods with ResNet50? Also, the baselines are outdated.
>
> A3: We've updated the comparison with recent methods in Table 2, and provided an "Accuracy vs. Complexity" comparison and trade-off figure in Appendix H.1 (Page 19). Please see the revised paper.
>
> As we investigated 2D CNN-based methods, most works focused on designing an extended module for temporal information learning based on ResNet-series baselines.
> Meanwhile, the latest 2D CNN-based methods, such as TAda [3] and TSQNet [4], also use ResNet-50 as the backbone. We thus consider ResNet-50 still commonly used in efficient video recognition comparison.
> Our work is focused on designing an efficient video recognition model, and hence we compare our method with 2D CNN-based methods with ResNet50.
>
> To further explore the impacts of backbones on 2D CNN methods, we use two stronger backbones (same complexity as ResNet50) searched by other training-free NAS methods (ZenNAS [1] 78.9% and MAEDET [2] 79.1% on ImageNet-1K) to build a strong baseline within the ResNet-like searching space.
> Results on the Sth-Sth V1 dataset are shown in the following table.
>
> | Method  | Frame  | GFLOPs  | Top1 | Top5|
> | :-- | --- | --- | ---| --- |
> | TSM  | 16 | 65  | 47.2 | 77.1 |
> | TSM + ZenNAS   | 16 | 69.8  | 47.9 | 77.5 |
> | TSM + MAEDET   | 16 | 69.6  | 48.1 | 78.1 |
> | TANet   | 16 | 66  | 47.6 | 77.7 |
> | TANet + ZenNAS   |16 | 70.8  | 48.1 | 78.2 |
> | TANet + MAEDET   | 16 | 70.5  | 48.2 | 78.0 |
> | E3D-M  | 16  | 4.7  | 49.4 | 78.1 |
> | E3D-L  | 16  | 18.3  | 51.1 | 78.7 |
>
> The above table indicates that the searched stronger backbone by training-free NAS methods can obtain better performance than the ResNet-50 backbone.
> Under fewer computational resources, E3D-M and E3D-L can achieve superior performance.
> In real-world applications, 2D CNN-based methods are widely used due to the better optimization of 2D convolutional operators in hardware. While hardware support for 3D convolutional operators is developing. In summary, 2D CNNs and 3D CNNs in efficient video recognition have their own advantages and defects, and our work focuses on developing an effective training-free NAS method for efficient 3D CNN video recognition.
>
> [1] Ming Lin et al. Zen-NAS: A zero-shot NAS for high-performance image recognition. ICCV 2021.
> [2] Zhenhong Sun et al. MAE-DET: Revisiting maximum entropy principle in zero-shot NAS for efficient object detection. ICML 2022.
> [3] Ziyuan Huang et al. TAda! Temporally-Adaptive Convolutions for Video Understanding. ICLR 2022.
> [4] Boyang Xia et al. Temporal Saliency Query Network for Efficient Video Recognition. ECCV 2022.
>
> ------
> We have updated the 2D CNN baseline with new literature and replaced ResNet-50 with the same computational complexity NAS backbones. In addition, we have provided a detailed comparison of "Accuracy vs. Complexity" in the revised paper. Please let us know if there are other works missing. We would be grateful if you could reconsider your score after taking into account the points above.

---

> > ### Comment · Reviewer_Jkpo · 2022-11-23
> > **Thanks for the responce**
> >
> > Thanks for the updated comparison. It solves most of my concerns and I would raise my score to 6: marginally above the acceptance threshold

---

### Official Review · Reviewer_SqC4 · 2022-10-26

**Confidence:** 2
**Correctness:** 3
**Technical Novelty And Significance:** 2
**Empirical Novelty And Significance:** 2
**Recommendation:** 5

**Clarity, Quality, Novelty And Reproducibility:**

Clarity: very clear
Quality: high
Novelty: Okay. Training free and entropy idea is explored on 2D, but haven't explore on 3D.
Originality: I didn't see similar works before.

**Strength And Weaknesses:**

Strength
1. The experiments are solid, showing a good alignment between the proxy task and the video recognition ability.
2. The results on Sth-Sth are good.

Question
1. “Table 9 shows three instantiations of E3D with varying complexity, including E3D-S (1.9G FLOPs), E3D-M (4.7G FLOPs), and E3D-L (18.3G FLOPs).” Wondering the details about how to get E3D-S/M/L – are they searched separately? Or search a model and then manually scale up.
2. Is it possible to get an even larger model (maybe the name would be E3D-XL)? If it is possible, would we expect higher performance?
3. Is it possible to extend this entropy idea to other architectures (eg transformer)?
4. Is it possible to trivially apply other training-free NAS on 3D task? I am a little bit concern that it is not convinced the proposed method is better than previous training-free NAS method.

Weakness
1. The results on Kinetics400 are not as good as the transformer based method (although I understand the transformers are computationally expensive)
2. See question 4: It seems mainly compared with manually designed architectures, but don't compare with other NAS methods.


**Summary Of The Paper:**

This paper aims to find a good 3D CNN architecture for video recognition. This study proposes to use a novel proxy task, maximizing a designed entropy value, to search for the effective architecture. The experiments show the effectiveness of the proposed algorithm.


**Summary Of The Review:**

This paper is clear. The story makes sense. The results of this paper is good, and the author shows that the proposed method is better than random search. However, it makes the paper less convincing if don't compare with other NAS based methods.

---

> ### Author Response · Authors · 2022-11-16
> **Response to Reviewer SqC4**
>
> Thanks for all your valuable comments. Please see below our response to the specific questions.
>
> > Q1: Table 9 shows three instantiations of E3D with varying complexity, including E3D-S (1.9G FLOPs), E3D-M (4.7G FLOPs), and E3D-L (18.3G FLOPs)." Wondering the details about how to get E3D-S/M/L – are they searched separately? Or search a model and then manually scale up.
>
> A1: All models are searched separately with different FLOPs budgets (1.9G, 4.7G, and 18.4G) for fair comparison with X3D-S/M/L as the baseline.
> Thanks for your reminder, we've revised the description of Section 3.4 and Appendix F to make it clear. Please see the revised version.
>
> > Q2: Is it possible to get an even larger model (maybe the name would be E3D-XL)? If it is possible, would we expect higher performance?
>
> A2: We try to get a larger model E3D-XL with a given 35.9G FLOPS budget (fair comparison with X3D-XL), the performance is Top-1 52.4% and Top-5 79.1% on the Sth-Sth V1 dataset, which is higher than E3D-L (Top-1 51.1%), as computational complexity increases.
> We will release our code and models in the camera-ready version for other researchers who want to use efficient E3D family or search for larger models.
>
> > Q3 & W1: Is it possible to extend this entropy idea to other architectures (eg transformer)?
>
> A3: As discussed in Appendix. I, we believe that the principle of maximum entropy is theoretically applicable to transformers. However, there exist some challenges to overcome. For example, Transformer has more complex components than CNN, such as 'Q' and 'K' kernel operation and multi-head attention, which is difficult to calculate the maximum entropy. In addition, the discrepancy of visual information in spatial and temporal dimensions by Transformer still remains a challenge. Thus, we leave Transformer-based networks as a future exploration.
>
> Although our E3D-L performs a little worse than transformer-based methods, E3D-L consumes much smaller computational complexity and parameters, which is in line with our claims that we propose to automatically design efficient 3D CNN architectures via a novel training-free neural architecture search approach tailored for 3D CNNs considering the model complexity.
>
> > Q4 & W2: Is it possible to trivially apply other training-free NAS on 3D task? I am a little bit concern that it is not convinced the proposed method is better than previous training-free NAS method.
>
> A4: Since there is no existing code available of training-free NAS methods for 3D CNNs, we provided the comparison on ImageNet-1K between our HomoEntr-Score searched 2D CNN architecture with other training-free methods (ZenNAS [1] and MAEDET [2]) in Appendix D.
>
> To apply these two approaches to video recognition,  we have now refined their implementations for 3D CNNs.
> Due to limited time and resources, we reproduced these two related training-free NAS methods and provide results on the Sth-Sth V1 dataset under the 1.9GFLOPs budget.
> The results are shown in the below table.
>
> | Training-free method   | GFLOPs | Search Devices|  Design Cost (hours)| Top-1 | Top-5|
> | :-- | ---| ---| ---| ---| ---|
> | X3D-S |  2 |  - |  - |  44.6 | 74.4 |
> | Zen-score [1]  |  1.9 |  GPU |  26 |  45.5 |  74.6 |
> | MAE-DET score [2]  | 1.9 |  GPU | 15  | 45.8 | 74.7|
> | E3D-S | 1.9 | CPU | 3  | 47.1 | 75.6 |
>
> According to the results in the above table, the performances of other training-free NAS methods, Zen-score [1] and MAE-DET score [2], are better than X3D-S, but the performance of our searched model E3D-S is higher. It indicates directly applying training-free NAS methods can be effective in the video recognition task, but it still needs spatio-temporal refinement on video understanding tasks, which our work mainly focuses on. We've added this comparison and detailed description in Appendix D of the revised paper, please see the revised paper.
>
> Besides, we compared our methods with a training-based NAS method, MoViNet, in Table 2, Table 3, Table 4, Table 5, and Table 11.
> These comparisons indicate that even though our method consumes extremely lower searching time than the training-based method, it can still obtain comparable results.
>
> [1] Ming Lin et al. Zen-NAS: A zero-shot NAS for high-performance image recognition. ICCV 2021.
> [2] Zhenhong Sun et al. MAE-DET: Revisiting maximum entropy principle in zero-shot NAS for efficient object detection. ICML 2022.
>
> ------
> We would be grateful if you could reconsider your score after taking into account the points above and addressing your concerns.

---

> > ### Comment · Reviewer_SqC4 · 2022-11-19
> > **Thanks for the response!**
> >
> > Most of my concerns are addressed. I think this paper overall makes sense, and has interesting results. I have no other concerns if Reviewer Jkpo is also happy with the baselines. Thanks!

---

### Official Review · Reviewer_RjKM · 2022-10-27

**Confidence:** 3
**Correctness:** 3
**Technical Novelty And Significance:** 3
**Empirical Novelty And Significance:** 3
**Recommendation:** 8

**Clarity, Quality, Novelty And Reproducibility:**

The paper is presented clearly. Although the search method is not new, however, the consideration of the feature maps, kernel size in different depths and refined by using the proposed Spatio-Temporal Entropy Score shows promising experimental results. I think it will provide useful insights to the community.

**Strength And Weaknesses:**

The strengths of the paper are:
+ This paper tried to exploit the important direction for the video action recognition, or generally the video intellengent analysis.
+ The authors proposed to estimate correlation between feature map and kernel size in different depths, and furthermore, they proposed the Spatio-Temporal Entropy Score.
+ The proposed STEntrScore is positively correlated with Top1 accuracy which indicates that the proposed spatio-temporal refinement can handle the discrepancy of visual information in spatial and temporal dimensions.
+ The performance on the benchmark datasets is promising for the research in this area.
+ Inference and training speed are appealing

The weaknesses of the paper are:
- No limitation discussion in the paper
- I would like to see some discussions on the affect of the temporal resolution on the performance, such as if we increase or decrease the number of frames we consider in the searching and final architecture, what the performance?



**Summary Of The Paper:**

This paper proposed a spatio-temporal entropy score (STEntr-Score) with a refinement factor to handle the discrepancy of visual information in spatial and temporal dimensions, through dynamically leveraging the correlation between the feature map size and kernel size depth-wisely. Then the proposed entropy-based 3D CNNs (E3D family), can then be efficiently searched by maximizing the STEntr-Score under a given computational budget, via the existing evolutionary algorithm without training the network parameters. Comprehensive comparisons on the popular benchmark datasets show the advances of the proposed method.

**Summary Of The Review:**

Overall, I think this is a good paper which will have positive contributions to this field.

---

> ### Author Response · Authors · 2022-11-16
> **Response to Reviewer RjKM**
>
> Thanks for all your positive comments. Please see below our response to the specific weakness.
>
> > W1: No limitation discussion in the paper.
>
> A1: We provided limitation discussion (named future direction) in Appendix I, including Data-driven design and transformer model.
> For data-driven design, If we replace the Gaussian input directly with target data, the output after a convolution will be random due to the Gaussian initialized weights, as the process of STEntr-Score based searching is contained without data training.
> For transformer-based model, there exist some challenges to overcome. For example, Transformer has more complex components than CNN, such as 'Q' and 'K' kernel operation and multi-head attention, which is difficult to calculate the maximum entropy.
> Although these challenges are difficult to overcome, these would be fascinating tasks for us in the future.
>
> Thanks for your reminder. We've changed the title of Appendix I to "Limitation Discussion", please see the revised paper.
>
> > W2: I would like to see some discussions on the affect of the temporal resolution on the performance, such as if we increase or decrease the number of frames we consider in the searching and final architecture, what the performance?
>
> A2: Thanks for your suggestion, we conducted a series of different temporal resolution experiments on Sth-Sth V1 dataset.
> (1). We simply scale the temporal resolution to train E3D-S architecture;
> (2). Under FLOPs in (1), we searched different models with different temporal resolutions (8, 16, 32) for each FLOPs budgets (1.2G, 2.3G and 4.7G);
> (3). Under the E3D-S budgets (1.9G), we searched different models with different temporal resolutions (8, 16, 32).
>
> The results are shown as below:
>
> | Model | Frames | Resolution | GFLOPs | Top1 | Top5 |
> | :-- | --- |  --- | --- | --- | --- |
> | E3D-S |  13  | 160  | 1.9  | 47.1 | 75.6 |
> |E3D-M | 16 | 160  | 4.7  | 49.4 |  78.1 |
> | Scaling temporal resolution in E3D-S |  8   | 160   | 1.2   | 43.5 |   72.8 |
> | | 16   | 160  | 2.3   | 47.5 | 76.5 |
> || 32   | 160  |  4.7  | 48.6 | 77.2 |
> |Searched Models with FLOPs in (1) as Budget | 8   | 160  | 1.2   |44.2 | 74.0 |
> || 16   |160  | 2.3   |48.1 |  77.1|
> || 32    |160  | 4.7   | 49.6 |  77.7 |
> | Searched Models  with E3D-S FLOPs as Budget | 8    |160   | 1.9   | 44.7 | 72.9 |
> ||16   | 160   | 1.9   | 47.7 | 76.2 |
> ||  32   | 160   |1.9   | 47.8 |  76.3|
>
> According to the result of the above table, we can obtain the following observations.
> (a) By scaling the frame number of the final E3D-S architecture from 8 to 16, the Top-1 accuracy increases as computational complexity increases from 1.2GFLOPs to 4.7GFLOPs.
> (b) The performances of searched models in (2) are higher than simply increasing the frame number of the final E3D-S architecture.
> (c) The performances of searched models in (3) obviously increase with the increase of the frame number until the frame value approaches 16.
> We consider the results are related to the characteristics of the Sth-Sth V1 dataset, which only includes 30-60 frames for each video. The results also indicate the effectiveness of our proposed STEntrScore that can estimate the spatio-temporal aggregation.
>
> Kindly note that the budgets and temporal resolution values of E3D-S/M/L are the same as X3D-S/M/L for fair comparison. Details are presented in Section 3.4 and Appendix F.

---

### Author Response · Authors · 2022-11-16
**General Response**

Dear reviewers and meta reviewers,

We appreciate all reviewers for their valuable comments and suggestions. We've thoroughly revised our manuscript by adding more details and ablation studies as follows:

* We have added a clear clarification of our searched  E3D-S/M/L budget settings (Section 3.4 and Appendix F);
* We have updated the Table 2 with a new 2D CNN-based method (CVPR 2022);
* We have added a comparison with other training-free NAS methods on 3D CNNs in Appendix D;
* We have added a accuracy/complexity trade-off comparison in Appendix H.1;
* We have revised the title of Appendix I as "Limitation Discussion".

The changes have been highlighted using blue font in the revised paper. We will release our code and models in the camera-ready version, and please see below our response to each reviewer. If you have any questions or suggestions, please put your comments on OpenReview.

---

### Decision · Program_Chairs · 2023-01-20

**Decision:**

Accept: poster

**Justification For Why Not Higher Score:**

Only one of the reviewer is enthusiastic about the work.

**Justification For Why Not Lower Score:**

The concerns of all reviewers have been addressed.

**Metareview: Summary, Strengths And Weaknesses:**

The paper proposes a network architecture search method for 3D video recognition network based on a spatial-temporal entropy metric. The results are strong. Though there are some concerns of the reviewers at the beginning, they are largely addressed after rebuttals provided by the authors (although one of the reviewers does not revise the score, he/she does not have concerns any more). Therefore, the meta-reviewer recommends to accept the work.

**Note From Pc:**

if the above contains the word "oral" or "spotlight" please see: "oral" presentation means -> notable-top-5% and "spotlight" means -> notable-top-25%. As stated in our emails, we are disassociating presentation type from AC recommendations